# GAS5 rs145204276 Ins/Del Polymorphism Is Associated with CRC Susceptibility in a Romanian Population [note 1]

**DOI:** 10.3390/ijms26073078

**Published:** 2025-03-27

**Authors:** Cecil Sorin Mirea, Michael Schenker, Bianca Petre-Mandache, Mihai-Gabriel Cucu, Georgiana-Cristiana Camen, Ionică Daniel Vîlcea, Bogdan Cristian Albu, Cosmin Vasile Obleagă, Mihai Călin Ciorbagiu, Ioana Streață, Răzvan Mihail Pleșea, Anca-Lelia Riza, Florin Burada

**Affiliations:** 1Department of Surgery, University of Medicine and Pharmacy of Craiova, 200349 Craiova, Romania; mirea.cecil@umfcv.ro (C.S.M.); ionica.vilcea@umfcv.ro (I.D.V.); cosmin.obleaga@umfcv.ro (C.V.O.); mihai.ciorbagiu@umfcv.ro (M.C.C.); 2Department of Surgery, Emergency Clinical County Hospital, 200642 Craiova, Romania; bogdanalbu9898@gmail.com; 3Department of Oncology, University of Medicine and Pharmacy of Craiova, 200349 Craiova, Romania; michael.schenker@umfcv.ro; 4Doctoral School, University of Medicine and Pharmacy of Craiova, 200349 Craiova, Romania; 5Laboratory of Human Genomics, University of Medicine and Pharmacy of Craiova, 200638 Craiova, Romania; ioana.streata@umfcv.ro (I.S.); razvan.plesea@umfcv.ro (R.M.P.); anca.costache@umfcv.ro (A.-L.R.); florin.burada@umfcv.ro (F.B.); 6Regional Centre of Medical Genetics Dolj, Emergency Clinical County Hospital Craiova, 200642 Craiova, Romania; 7Department of Radiology and Medical Imaging, University of Medicine and Pharmacy of Craiova, 200349 Craiova, Romania; georgiana.camen@umfcv.ro

**Keywords:** colorectal cancer, long non-coding RNAs, gene, single nucleotide polymorphism, genotype

## Abstract

Colorectal cancer (CRC) is a leading cause of cancer-related morbidity and mortality, influenced by both genetic and epigenetic factors. Long non-coding RNAs (lncRNAs) such as GAS5 and CASC8 have been implicated in cancer susceptibility. This study aimed to assess the association of GAS5 rs145204276 ins/del and CASC8 rs10505477 A>G polymorphisms with CRC risk in a Romanian population. A case-control study was conducted, including 156 CRC patients and 195 healthy controls. Genotyping for GAS5 and CASC8 polymorphisms was performed using real-time PCR, and the association with CRC risk was evaluated using logistic regression to calculate odds ratios (OR) and 95% confidence intervals (CI). The carriers of GAS5 rs145204276 del allele was significantly associated with increased CRC risk (OR: 2.13, 95% CI: 1.24–3.63, *p* = 0.005) in a dominant model. In the subgroup analysis, the association of GAS5 rs145204276 ins/del polymorphism was restricted to distal colon cancer cases (OR: 2.98, 95% CI: 1.57–5.66, *p* = 0.001), advanced tumor stages (III + IV) (OR: 2.54, 95% CI: 1.31–4.91, *p* = 0.007), and poorly differentiated tumors (G3) (OR: 3.98, 95% CI: 1.49–10.59, *p* = 0.009). No significant correlation was found for the CASC8 rs10505477 A>G polymorphism. GAS5 rs145204276 polymorphism may influence CRC susceptibility, particularly in distal tumors and advanced stages. However, CASC8 rs10505477 polymorphism showed no association with CRC risk in this Romanian cohort.

## 1. Introduction

Colorectal cancer (CRC) is a major public health concern, ranking as the third most diagnosed cancer and the second cause of cancer-related deaths worldwide. CRC accounts for approximately 1.9 million new cases and 935,000 deaths annually worldwide, thus heavily impacting healthcare systems [1,2]. Although CRC primarily affects individuals over 50 years, recent studies indicate an alarming increase in incidence among younger adults under 50 years [3]. In addition to genetic and epigenetic changes [3,4,5], several environmental factors [4,5,6,7,8] have been found to increase the risk of CRC.

Long noncoding RNAs (lncRNAs) have emerged as critical regulators in the etiology of CRC. LncRNAs are transcripts longer than 200 nucleotides with no or limited protein-coding potential and have been reported to exert regulatory effects on cellular processes such as proliferation, apoptosis, and metastasis [6,7,8,9]. LncRNAs can influence CRC cell proliferation through various mechanisms, for example, some lncRNAs act as competing endogenous RNAs (ceRNAs), sequestering microRNAs (miRNAs) and preventing them from inhibiting their target mRNAs. This results in the overexpression of oncogenes promoting cellular proliferation. Additionally, certain lncRNAs interact with chromatin-modifying complexes or transcription factors to enhance gene expression, thus driving the cell cycle forward [10,11]. One characteristic of cancer growth linked to the deregulation of lncRNAs in CRC is the avoidance of apoptosis. By modulating the expression of pro-apoptotic and anti-apoptotic genes, various lncRNAs inhibit apoptotic pathways. The overexpression of certain oncogenic lncRNAs can suppress the caspases’ activation, the central executors of apoptosis, thereby enabling cancer cells to proliferate uncontrollably and survive [12]. LncRNAs engage in CRC metastasis by regulating the epithelial-mesenchymal transition (EMT), increasing their invasive and migratory capacities. Some lncRNAs have been shown to promote EMT by interacting with signaling pathways such as Wnt/β-catenin. This interaction promotes the upregulation of mesenchymal markers and downregulation of epithelial markers, therefore aiding in the spread of cancer cells from the primary tumor to remote locations [11,13,14,15].

One notable lncRNA, *Growth Arrest-Specific 5* (*GAS5*), has been identified as a tumor suppressor in various cancers, including CRC [16,17,18,19,20,21]. *GAS5* polymorphisms, particularly rs145204276 (a 5-bp ins/del in its promoter region), affect the gene’s transcriptional activity and are associated with cancer susceptibility [22,23,24]. The deletion (del) allele of rs145204276 has been associated with a protective effect against CRC but also with increasing the risk for other types of cancer, emphasizing its complex role in tumorigenesis [25,26,27].

Another lncRNA of interest in CRC is *Cancer Susceptibility Candidate 8* (*CASC8*), located within the region of 8q24. *CASC8* plays a role in tumor progression by interacting with chromatin-modifying complexes and transcription factors. The rs10505477 polymorphism in the *CASC8* intron is one of the most studied single nucleotide polymorphisms (SNPs), with studies indicating its role in CRC susceptibility [28,29,30]. In addition, *CASC8* dysregulation was present in numerous cancers, including CRC, prostate cancer, and gastric cancer [28,29,31].

In addition to *GAS5* and *CASC8*, other lncRNAs have been implicated in cancer development and progression. For instance, *MALAT1* has been the subject of extensive research across multiple cancer types, including CRC [32,33,34,35]. Furthermore, *HOTAIR* has been identified as a crucial lncRNA in tumorigenesis and functions as a scaffold for chromatin-modifying complexes that assist with the epigenetic silencing of tumor suppressor genes. Its overexpression in cancers such as CRC is linked with poor prognosis [36,37,38]. *Prostate Cancer Non-Coding RNA 1* (*PRNCR1*) was implicated predominantly in prostate cancer, where it modulates androgen receptor activity and induces tumor growth, recent findings show that *PRNCR1* is also involved in CRC [39,40,41].

The carcinogenesis of CRC is a multistep process involving the accumulation of genetic mutations and epigenetic modifications, involving lncRNAs like *GAS5* and *CASC8* as important modulators in key signaling pathways like the Wnt/β-catenin and Hippo/YAP, which play a major role in tumor initiation and progression [42,43,44]. *GAS5* was found to suppress CRC cell proliferation and induce apoptosis [45,46], whereas *CASC8* plays a role in transcriptional regulation through its enhancer-like activity [47,48].

Genetic susceptibility to multifactorial diseases, including CRC, varies across geographic regions and ethnic groups. While numerous studies have investigated the role of SNPs in CRC risk, most data on *GAS5* rs145204276 and *CASC8* rs10505477 polymorphisms come from Asian populations, with limited research in European cohorts. In this respect, the aim of our study was to examine the relationship between the *GAS5* rs145204276 and *CASC8* rs10505477 polymorphisms and the susceptibility to CRC through a case-control study in a Romanian population (an Eastern European population), an ethnic group in which the association between these polymorphisms and CRC susceptibility has not previously been studied.

## 2. Results

We genotyped two lncRNA polymorphisms (*GAS5* rs145204276 and *CASC8* rs10505477) in a total of 351 subjects, including 156 CRC patients and 195 healthy controls. The groups were matched for age and sex, with no statistically significant differences between CRC patients and the controls (*p* > 0.05). The parameters and clinical characteristics of the CRC patients are presented in Table 1. The mean age was 68.61 ± 9 years for CRC patients and 66.69 ± 7.94 years for the controls. Among the CRC group, 89 were males and 67 were females. Tumor localization was as follows: 35 cases in the proximal colon, 69 in the distal colon, and 52 in the rectum.

Based on Dukes’ classification, 89 patients were classified as Stages A + B, while 67 were in Stages C + D. Tumor differentiation grades revealed 23 cases as well-differentiated (G1), 113 as moderately differentiated (G2), and 20 as poorly differentiated (G3). For each polymorphism analyzed, the genotype distributions were consistent with the expectations of Hardy–Weinberg equilibrium in the case and control groups (*p* > 0.05). This indicates no evidence of genetic drift or selective advantage within the case and control population, supporting the reliability of the genotyping process and the representativeness of both the case and control groups. (Table 2). 

### 2.1. LncRNA Polymorphisms and CRC Susceptibility

For each polymorphism, the minor allele was considered the risk variant in comparison to the wild-type allele. The genotype and allele frequencies, along with their association with CRC risk, are presented in Table 3.

A significant association was observed for the *GAS5* rs145204276 ins/del polymorphism in the codominant, dominant, and allelic genetic models. Individuals with the ins/del genotype showed a higher frequency among CRC patients compared to controls. The strongest association was identified in the dominant model, where the relative risk for carriers of the del allele was significantly elevated (OR: 2.13, 95% CI: 1.24–3.63, *p* = 0.005) compared to individuals with the wild-type genotype (ins/ins). Furthermore, a significant correlation was observed in the allelic model, when del allele was compared with the more frequent ins allele (OR: 1.99, 95% CI: 1.21–3.27, *p* = 0.006).

No significant association was found for the *CASC8* rs10505477 A>G SNP in any of the genetic models (codominant, dominant, recessive, or allelic). The distribution of *CASC8* genotypes and allele frequencies did not differ significantly between CRC patients and controls (*p* > 0.025).

### 2.2. Association of LncRNA Polymorphisms with Tumor Stage and Histological Grade

The association of lncRNA polymorphisms with tumor stage and histological grade was evaluated, with results summarized in Table 4 and Table 5. For *GAS5* rs145204276, significant associations were observed between the del allele and advanced tumor stages (III + IV) (*p* = 0.015) as well as poorly differentiated tumors (G3) (*p* = 0.009). In contrast, no correlations were found between *CASC8* rs10505477 A>G SNP and tumor stage, and histological grade.

### 2.3. LncRNA Polymorphisms and Tumor Localization

Further analysis of tumor site revealed a strong association between the *GAS5* rs145204276 del allele and tumors located in the distal colon in the dominant model, with the relative risk for del carriers being 2.98 (95% CI: 1.57–5.66) compared to the more frequent ins/ins genotype. No associations were found for *CASC8* rs10505477 A>G SNP relating tumor localization, including proximal versus distal CRC and rectal cancer subgroups (*p* > 0.025) (Table 6).

#### Post Hoc Power Analysis

To evaluate the robustness of our findings, a post hoc power analysis was performed based on the observed effect sizes in our study. With a sample size of 156 CRC patients and 195 healthy controls, the analysis demonstrated that our study had approximately 80% power to detect an effect size of 0.32 at a significance level of 0.05. This indicates that the sample size was adequate to identify meaningful associations between the studied polymorphisms and CRC susceptibility.

## 3. Discussion

In this association case-control study, we evaluated the correlation between two lncRNA polymorphisms and their potential influence on CRC susceptibility in a Romanian population and we detected an association for *GAS5* rs145204276 but not for *CASC8* rs10505477 [49]. These polymorphisms were selected based on their location in functional regions of lncRNA genes and their previously reported roles in modulating cancer risk [10,50,51,52,53]. By examining their distribution and association with CRC, we aimed to determine the genetic contribution of these lncRNAs to CRC pathogenesis in this specific population.

Several case-control studies have explored the association between lncRNA polymorphisms and susceptibility to CRC, some of which reported significant associations while others reported no clear link.

We found a positive association between the *GAS5* rs145204276 deletion (del) allele and CRC susceptibility. In our study, the del allele was associated with an increased risk of CRC [49], suggesting that it may act as a risk factor rather than providing a protective effect. This finding contrasts with some previous evidence suggesting a tumor-suppressive role of GAS5 in certain cancers, pointing towards the complexity of lncRNA function in different tissues and disease conditions. Further, we observed no significant association between the *CASC8* rs10505477 and CRC susceptibility [49], suggesting that this polymorphism may not play a major role in CRC risk within the studied population.

*GAS5*, a well-documented lncRNA with tumor-suppressive potential, is involved in key cellular processes such as apoptosis and proliferation [22,50,54,55,56]. *GAS5* exerts its function through interactions with critical cellular pathways, including the p53 pathway and microRNAs, such as miR-21 [57,58,59,60]. *GAS5* can act as a molecular sponge for various microRNAs (miRNAs), thereby modulating the expression of miRNA target genes involved in tumorigenesis. The rs145204276 polymorphism may influence the binding affinity between *GAS5* and specific miRNAs, altering the regulatory networks that control cell proliferation and apoptosis. Although some previous reports have linked increased *GAS5* expression to reduced cancer risk [6,50,61,62], our findings suggest that the rs145204276 del allele may enhance CRC risk in Romanian population, mainly for advanced and poorly differentiated tumors. This observation aligns with other studies reporting that *GAS5* expression can have varying effects regarding metastasis, prognosis, or proliferation [6,63], also depending on the cancer type and tissue-specific regulatory mechanisms [64,65]. In a study by Tao et al., the del allele was associated with increased hepatocellular carcinoma risk due to altered *GAS5* transcript activity, indicative of tissue-specific gene regulatory mechanisms [66]. Similarly, *GAS5* polymorphisms have been linked to glioma progression, where *GAS5* expression was dysregulated, resulting in impaired tumor suppression [67]. Such variations may result from tissue-specific regulatory mechanisms and interactions with distinct molecular pathways in different cancer types.

*CASC8*, located in the 8q24 chromosomal region, has also been studied for its association with cancer susceptibility, particularly in CRC. This locus is well known for harboring risk variants that regulate key oncogenic pathways, including the Wnt/β-catenin signaling pathway and the *MYC* oncogene, both of which play critical roles in CRC initiation and progression [28,68]. Our study did not show a statistically significant correlation in this Romanian population [49], indicating that this polymorphism may not be a strong risk factor for CRC in our cohort. While some studies linked *CASC8* rs10505477 to elevated CRC risk [69,70,71,72,73], conflicting results have been reported in other populations [74,75]. For instance, Yao et al. found that rs10505477 was associated with the A allele in relation to CRC risk in certain ethnic groups, indicating a potential role for genetic background and environmental interactions in modulating *CASC8*-associated cancer susceptibility [74]. Interestingly, *CASC8* polymorphisms have also been associated with differential cancer risks in adenomas and other cancer types. A case-control study by Gargallo et al. highlighted that *CASC8* rs10505477 polymorphisms were associated with a lower risk of colorectal adenomas, suggesting that this locus may play varying roles depending on cancer subtype or disease stage [75].

There are several limitations in our study that should be noted. First, the sample size was relatively modest, which may have limited the power to detect subtle associations, particularly in subgroup analyses based on tumor location and disease stage. Second, we focused on only two lncRNA polymorphisms (*GAS5* rs145204276 and *CASC8* rs10505477) and did not include other potentially relevant genetic variants. Third, our study lacked functional assays to validate the biological implications of these polymorphisms, such as their effects on *GAS5* and *CASC8* expression or activity. Additionally, we did not account for environmental factors, including dietary habits, lifestyle, or microbiome composition, which are known to influence CRC risk. Lastly, since our study population consisted primarily of individuals of Romanian ethnicity, the findings may not be generalizable to other populations, necessitating further studies in diverse cohorts.

## 4. Materials and Methods

### 4.1. Subjects

A total of 351 individuals were included in this study, comprising of 156 patients diagnosed with CRC and 195 healthy controls. Genotyping was conducted for the *GAS5* rs145204276 and *CASC8* rs10505477 polymorphisms.

Patients included in the CRC group were diagnosed with sporadic colorectal adenocarcinoma, confirmed through histopathological analysis. All CRC cases were verified as having no prior history of hereditary cancer syndromes, such as familial adenomatous polyposis or Lynch syndrome (hereditary non-polyposis CRC).

Patients were excluded from the case group if they presented with metastatic colorectal adenocarcinoma originating from a primary malignancy in another organ. Additionally, individuals with a documented history of other primary malignancies were excluded to ensure the focus remained on sporadic CRC.

The control group consisted of cancer-free individuals with no personal history of CRC or other malignancies. Controls were matched to the CRC group by age and gender to ensure the comparability of the groups.

This study adhered to the ethical principles outlined in the Declaration of Helsinki https://www.wma.net/what-we-do/medical-ethics/declaration-of-helsinki/ (accessed on 24 January 2019), and was approved by the Ethics Committee of the University of Medicine and Pharmacy of Craiova, Romania (No. 39/18 February 2019).

### 4.2. Genotyping

Blood samples were collected from each participant in 6 mL EDTA tubes and stored at 4 °C until processing. Genomic DNA was extracted from peripheral blood leukocytes using the Wizard^®^ Genomic DNA Purification Kit (Promega, Madison, WI, USA), following the manufacturer’s instructions. After extraction, DNA quantity and quality were assessed to ensure a minimum concentration of 20 ng. Absorbance readings at 260 nm and 320 nm were used to evaluate contamination, with a target absorbance ratio of at least 1.7. DNA normalization was performed to ensure uniform starting material for downstream analyses.

The *GAS5* rs145204276 (ins/del) and *CASC8* rs10505477 (A>G) polymorphisms were genotyped using real-time polymerase chain reaction (RT-PCR) with TaqMan predesigned probes. RT-PCR was conducted in 384-well plates on a ViiA™ 7 RT-PCR System (Life Technologies, Carlsbad, CA, USA). Before processing all samples, amplification conditions for the probes were optimized using random samples. Each plate included at least two negative controls (no DNA) to monitor for contamination.

Genotyping was performed in a reaction volume of 5 µL, comprising 2.5 µL reaction master mix, 1.125 µL ultrapure water, 0.250 µL TaqMan probe mix, and 1.125 µL of DNA extract. The thermal cycling conditions included an initial hold at 60 °C for 30 s and 95 °C for 10 min, followed by 40 cycles of 95 °C for 15 s and 60 °C for 1 min. Genotypes were determined by fluorescence detection using FAM and VIC dye labeling, with ROX used as a passive reference dye. All genotyping was conducted in a blinded manner, without knowledge of participant phenotype.

### 4.3. Statistical Analysis

The Hardy–Weinberg equilibrium (HWE) of the *GAS5* rs145204276 and *CASC8* rs10505477 polymorphisms was assessed using the goodness-of-fit chi-square (χ^2^) test. Logistic regression analysis was employed to evaluate the association between these polymorphisms and CRC susceptibility. The analysis was adjusted for potential confounders, including age and gender, to mitigate their impact on CRC risk. Odds ratios (ORs) and 95% confidence intervals (CIs) were calculated to quantify the strength of the association under codominant, dominant, recessive, and allelic models.

The codominant model examined the risk of CRC for each genotype separately (wild-type, heterozygous, and mutant). The dominant model compared individuals carrying at least one mutant allele (heterozygous or mutant) to those with the wild-type genotype. The recessive model assessed the risk associated with carrying two mutant alleles compared to those with one or no mutant alleles. The allelic model evaluated the frequency of mutant versus wild-type alleles in cases and controls. These models provided a comprehensive analysis of the relationship between the *GAS5* rs145204276 and *CASC8* rs10505477 polymorphisms and CRC susceptibility under various genetic inheritance patterns.

Statistical analyses were performed using Microsoft Excel 2019 (MSO Version 2409) and SPSS Statistics for Windows, Version 22.0 (IBM SPSS Statistics, Armonk, NY, USA). The Bonferroni-corrected alpha level was set at 0.025 (0.05/2 SNPs)

Baseline characteristics of the CRC and control groups were compared using appropriate statistical tests based on the type of variable. For categorical variables, such as gender distribution, the chi-square (χ^2^) test was applied to determine whether there were significant differences between groups. For continuous variables, such as age, the independent *t*-test was used to compare the means between cases and controls. A *p*-value of less than 0.025 was considered statistically significant for all analyses. The minor allele frequency (MAF) for each polymorphism was calculated in both the case and control groups.

## 5. Conclusions

In conclusion, our study revealed that the *GAS5* rs145204276 del allele is associated with an increased risk of CRC, mainly for advanced and poorly differentiated tumors, suggesting its role as a potential risk factor, while no significant association was observed for the *CASC8* rs10505477 polymorphism. These findings highlight the complex and context-dependent roles of lncRNAs in cancer susceptibility. Future research involving larger, multi-ethnic cohorts and functional analyses are necessary to further explore the role of these polymorphisms and clarify the underlying molecular mechanisms.

## Figures and Tables

**Table 1 ijms-26-03078-t001:** Baseline characteristics of study subjects.

Variable	Colorectal Cancer	Control
*N*	156	195
Male/female	89/67	143/90
Age (years), mean ± SD	68.61 ± 9	66.69 ± 7.94
Location		
- proximal	35	
- distal	69	
- rectum	52	
Tumor stage—Dukes stage		
- A + B	89	
- C + D	67	
Differentiation grade		
- G1—well	23	
- G2—moderate	113	
- G3—poor	20	

**Table 2 ijms-26-03078-t002:** Minor allele frequencies and Hardy–Weinberg equilibrium values in the control group.

Polymorphism	MAF	χ^2^	*p*
Control			
*GAS5* rs145204276 ins/del (CAAGG>-)	0.07	0.007	0.93
*CASC8* rs10505477 A>G	0.47	0.09	0.76
Cases			
*GAS5* rs145204276 ins/del (CAAGG>-)	0.14	0.42	0.52
*CASC8* rs10505477 A>G	0.46	0.56	0.45

**Table 3 ijms-26-03078-t003:** Association between lncRNA polymorphisms and CRC under multiple models of inheritance.

Polymorphism	Colorectal Cancer (*n* = 156)	Control (*n* = 195)	OR (95% CI)	*p* Value
*GAS5* rs145204276 ins/del (CAAGG>-)				
Codominant				
ins/ins	115 (73.72%)	167 (85.64%)	Reference	-
ins/del	39 (25.00%)	27 (13.85%)	2.09 (1.22–3.62)	0.007
del/del	2 (1.28%)	1 (0.51%)	2.90 (0.26–32.41)	0.36
Dominant				
ins/ins	115 (73.72%)	167 (85.64%)	Reference	-
del/del + ins/del	41 (26.28%)	28 (14.36%)	2.13 (1.24–3.63)	0.005
Recessive				
ins/ins + ins/del	154 (98.72%)	194 (99.49%)	Reference	-
del/del	2 (1.28%)	1 (0.51%)	0.39 (0.03–4.42)	0.437
Allelic				
ins	269 (86.22%)	361 (92.56%)	Reference	-
del	43 (13.78%)	29 (7.44%)	1.99 (1.21–3.27)	0.006
*CASC8* rs10505477 A>G				
Codominant				
AA	30 (19.23%)	44 (22.56%)	Reference	-
AG	82 (52.56%)	95 (48.72%)	1.27 (0.73–2.19)	0.40
GG	44 (28.21%)	56 (28.72%)	1.15 (0.63–2.12)	0.65
Dominant				
AA	30 (19.23%)	44 (22.56%)	Reference	-
GG + AG	126 (80.77%)	151 (77.44%)	1.22 (0.73–2.06)	0.45
Recessive				
AA + AG	112 (71.79%)	139 (71.28%)	Reference	-
GG	44 (28.21%)	56 (28.72%)	0.97 (0.61–1.56)	0.92
Allelic				
A	142 (45.51%)	183 (46.92%)	Reference	-
G	170 (54.49%)	207 (53.08%)	1.06 (0.79–1.43)	0.71

**Table 4 ijms-26-03078-t004:** Association between lncRNA polymorphisms and TNM stage of colorectal cancer.

Polymorphism	Tumor Stage I + II (A + B = 89) (%)	OR (95% CI); *p*	Tumor Stage III + IV (C + D = 67) (*n* %)	OR (95% CI); *p*
*GAS5* rs145204276 ins/del (CAAGG>-)				
ins/ins	68 (76.40%)	Reference	47 (70.15%)	Reference
ins/del	21 (23.60%)	1.91 (1.01–3.61); 0.049	18 (26.87%)	2.37 (1.20–4.67); 0.0015
del/del	0 (0%)	/	2 (2.98%)	7.11 (0.63–80.1); 0.101
del carriers	21 (23.60%)	1.84 (0.98–3.47); 0.06	20 (29.85%)	2.54 (1.31–4.91); 0.007
*CASC8* rs10505477 A>G				
AA	16 (17.98%)	Reference	14 (20.90%)	Reference
AG	50 (56.18%)	1.45 (0.74–2.82); 0.27	32 (47.76%)	1.06 (0.51–2.18); 0.88
GG	23 (25.84%)	1.13 (0.53–2.39); 0.75	21 (31.34%)	1.18 (0.54–2.58); 0.68
G carriers	73 (82.02%)	1.33 (0.70–2.51); 0.37	53 (79.10%)	1.10 (0.56–2.17); 0.78

**Table 5 ijms-26-03078-t005:** Association between lncRNA polymorphisms and colorectal cancer in the histologic grade subgroups.

	G1			G2			G3		
SNP	*N* = 23	OR (95% CI)	*p*	*N* = 113	OR (95% CI)	*p*	*N* = 20	OR (95% CI)	*p*
*GAS5* rs145204276									
ins/ins	16	Reference		87	Reference		12	Reference	0.007
ins/del	6	2.32 (0.83–6.45)	0.125	25	1.78 (0.97–3.25)	0.06	8	4.12 (1.54–11.01)	0.007
del/del	1	10.44 (0.62–174.9)	0.13	1	1.92 (0.12–31.06)	0.65	0	/	/
del carriers	7	2.61 (0.98–6.91)	0.07	26	1.78 (0.98–3.23)	0.06	8	3.98 (1.49–10.59)	0.009
*CASC8* rs10505477									
AA	3	Reference		24	Reference		3	Reference	
AG	13	2.01 (0.54–7.40)	0.27	61	1.18 (0.65–2.13)	0.59	8	1.24 (0.31–4.88)	0.76
GG	7	1.83 (0.45–7.50)	0.39	28	0.92 (0.47–1.80)	0.80	9	2.36 (0.60–9.23)	0.19
G carriers	20	1.94 (0.55–6.84)	0.27	89	1.08 (0.62–1.89)	0.79	17	1.65 (0.46–5.89)	0.42

**Table 6 ijms-26-03078-t006:** Association between lncRNA polymorphisms and colorectal cancer in the tumor site subgroups.

Polymorphism	Proximal CRC		Distal CRC		Rectal CRC	
	*N* = 35	OR (95% CI); *p*	*N* = 69	OR (95% CI); *p*	*N* = 52	OR (95% CI); *p*
*GAS5* rs145204276						
ins/ins	29	Reference	46	Reference	40	Reference
ins/del	5	1.07 (0.38–2.99); 0.90	22	2.96 (1.54–5.67); 0.001	12	1.86 (0.87–3.98); 0.12
del/del	1	5.76 (0.35–94.68); 0.25	1	3.63 (0.22–59.17); 0.38	0	/
del carriers	6	1.23 (0.47–3.24); 0.67	23	2.98 (1.57–5.66); 0.001	12	1.79 (0.84–3.82); 0.14
*CASC8* rs10505477						
AA	7	Reference	13	Reference	10	Reference
AG	19	1.26 (0.49–3.21); 0.63	37	1.32 (0.64–2.73); 0.45	26	1.20 (0.53–2.71); 0.65
GG	9	1.01 (0.35–2.93); 0.98	19	1.15 (0.51–2.56); 0.74	16	1.26 (0.52–3.04); 0.61
G carriers	28	1.17 (0.48–2.85); 0.73	56	1.26 (0.63–2.50); 0.51	42	1.22 (0.57–2.64); 0.60

## Data Availability

All data presented here are available from the authors upon reasonable request.

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
