# Peer review of "GAS5 rs145204276 Ins/Del Polymorphism Is Associated with CRC Susceptibility in a Romanian Population [Author-notes fn1-ijms-26-03078]"

_ijms, 2025, doi:10.3390/ijms26073078_

Round 1
Reviewer 1 Report
Comments and Suggestions for Authors
The manuscript investigates the association between GAS5 and CASC8 gene polymorphisms and colorectal cancer (CRC) susceptibility in the Romanian population. Genotyping was performed using real-time PCR and TaqMan probes, ensuring methodological reliability. This study provides novel population-specific data on lncRNA polymorphisms and CRC susceptibility, demonstrating reasonable methodology and some degree of innovation in its findings. However, improvements are needed in statistical analysis, data presentation, and the depth of discussion, particularly in addressing study limitations.
- The manuscript does not mention adjustments for multiple comparisons (e.g., Bonferroni correction or FDR) in cases of multiple genetic models and subgroup analyses. Although the main finding (GAS5 rs145204276, p = 0.005) is statistically significant, clarification on whether correction was applied is necessary to mitigate the risk of false positives. Additionally, further discussion on the potential mechanisms of GAS5 polymorphism in different tumor locations and stages is recommended.
- The G3 group consists of only 20 samples, which may affect result robustness. It is advisable to explicitly acknowledge the limitations of subgroup analysis in the discussion.
- Some references lack page numbers.
- Table 5 should be reformatted for clarity.
- The HWE test results for the case group should be provided to rule out potential selection bias.
Author Response
Dear Reviewer,
We are grateful for the amount of time and effort that you dedicated to providing feedback on our manuscript and are thankful for the insightful comments and valuable improvements to our paper. We responded point-by-point to all comments raised and the changes can be found in the track-changes revised manuscript. We hope to have successfully addressed your concerns below.
We also have expanded the main text of the manuscript by including more detailed information according to your suggestions, as well as updating the reference list.
Best wishes,
The authors
Comments and Suggestions for Authors
The manuscript investigates the association between GAS5 and CASC8 gene polymorphisms and colorectal cancer (CRC) susceptibility in the Romanian population. Genotyping was performed using real-time PCR and TaqMan probes, ensuring methodological reliability. This study provides novel population-specific data on lncRNA polymorphisms and CRC susceptibility, demonstrating reasonable methodology and some degree of innovation in its findings. However, improvements are needed in statistical analysis, data presentation, and the depth of discussion, particularly in addressing study limitations.
- The manuscript does not mention adjustments for multiple comparisons (e.g., Bonferroni correction or FDR) in cases of multiple genetic models and subgroup analyses. Although the main finding (GAS5 rs145204276, p = 0.005) is statistically significant, clarification on whether correction was applied is necessary to mitigate the risk of false positives. Additionally, further discussion on the potential mechanisms of GAS5 polymorphism in different tumor locations and stages is recommended.
Response:
We appreciate your insightful feedback. We have now incorporated the Bonferroni correction into our analysis, adjusting the alpha level to 0.025 (0.05/2 SNPs). The revised manuscript also provides further details regarding the mechanisms by which GAS5 polymorphisms may influence tumor characteristics across various locations and stages.
- The G3 group consists of only 20 samples, which may affect result robustness. It is advisable to explicitly acknowledge the limitations of subgroup analysis in the discussion.
Response:
Certainly, the G3 subgroup (n = 20) is small, and it was included in the limitations of the study. Further studies on larger cohorts are required.
- Some references lack page numbers.
Response:
We have updated the references. Some references are missing pagination, and we adhered to the journal's guidelines for citation.
- Table 5 should be reformatted for clarity.
Response:
We have reformatted the table, it seemed to be an incompatibility issue with MS. Word.
- The HWE test results for the case group should be provided to rule out potential selection bias.
Response:
We have addressed this issue, and Table 2 now includes the HWE test results for the case group.

Reviewer 2 Report
Comments and Suggestions for Authors
Remarks:
- How do the authors guarantee susceptibility to CRC with a small n?
- Were environmental factors and diet not evaluated?
- With the findings presented, can it be said that there is susceptibility?
- What is the justification for using the Romanian population? Make this clear in the text
Comments on the Quality of English Language- The text needs minor grammatical and text flow corrections to improve connectivity and fluidity for the reader
Author Response
Dear Reviewer,
We are grateful for the amount of time and effort that you dedicated to providing feedback on our manuscript and are thankful for the insightful comments and valuable improvements to our paper. We responded point-by-point to all comments raised and the changes can be found in the track-changes revised manuscript. We hope to have successfully addressed your concerns below.
We also have expanded the main text of the manuscript by including more detailed information according to your suggestions.
Best wishes,
The authors
Comments and Suggestions for Authors
Remarks:
- How do the authors guarantee susceptibility to CRC with a small n?
Response:
Thank you for your comment. Our study suggests that GAS5 rs145204276 ins/del polymorphism could influence CRC susceptibility in a Romanian population. However, as we stated in the conclusion, larger studies on different ethnic groups are required in order to clarify the role of these SNPs in CRC. We performed a post hoc power analysis, showing that our study had approximately 80% power to detect an effect size of 0.32 at a significance level of 0.05.
- Were environmental factors and diet not evaluated?
Response:
Environmental and dietary factors were not analyzed in this study. We addressed this issue in the limitation section.
- With the findings presented, can it be said that there is susceptibility?
Response:
In this study, we found that the GAS5 rs145204276 del allele is associated with an increased risk of CRC in our cohort. This SNP may particularly affect CRC susceptibility in cases of advanced-stage and poorly differentiated tumors. However, no significant association was identified for the CASC8 rs10505477 polymorphism. Additionally, larger studies involving diverse ethnic groups are necessary to further elucidate the role of these SNPs in CRC.
- What is the justification for using the Romanian population? Make this clear in the text
Response:
A significant body of data concerning SNP distribution and its association with various multifactorial diseases, including colorectal cancer, highlights notable differences across geographic regions and ethnic groups. This study aims to investigate, for the first time in an Eastern European population, the association between the GAS5 rs145204276 and CASC8 rs10505477 polymorphisms and the susceptibility to CRC, a case-control design. Furthermore, genetic studies focusing on these SNPs and their link to CRC susceptibility in European populations remain limited, with most findings predominantly reported in Asian cohorts.
We have rephrased some sentences to make them more clear and concise.
Comments on the Quality of English Language
- The text needs minor grammatical and text flow corrections to improve connectivity and fluidity for the reader
Response:
We appreciate the time and effort spent on reviewing our manuscript; your valuable feedback has been instrumental. We have made several changes to enhance the connectivity and fluidity for readers.
